# In Situ X-ray Synchrotron Radiation Analysis, Tensile- and Biodegradation Testing of Redox-Alloyed and Sintered MgCa-Alloy Parts Produced by Metal Injection Moulding

Martin Wolff *, Heike Helmholz, Monika Luczak, Daniel Strerath, Thomas Ebel and Regine Willumeit-Römer

Helmholtz-Zentrum Hereon, Institute of Metallic Biomaterials, D-21502 Geesthacht, Germany; heike.helmholz@hereon.de (H.H.); monika.luczak@hereon.de (M.L.); daniel.strerath@hereon.de (D.S.); thomas.ebel@hereon.de (T.E.); regine.willumeit@hereon.de (R.W.-R.)
* Correspondence: martin.wolff@hereon.de; Tel.: +49-4152-871916

**Abstract:** Binary MgCa alloys are one of the promising and well investigated biodegradable metals and therefore a good standard for the study of novel processing routes. In this investigation, novel powder metallurgical (PM) blending and sintering methods were applied for the generation of biodegradable MgCa test specimens, using metal injection moulding (MIM). In addition to the classical PM-blending route using Ca-containing master-alloys, Ca-containing ceramics and hydrides, as there are CaO and $CaH_2$, were used separately and in a stoichiometric mixture. In situ X-ray synchrotron radiation experiments were performed for a deeper understanding of alloy forming mechanisms during sintering. Mechanical and degradation performance was investigated by tensile testing and the monitoring of biodegradation under physiological conditions. Besides its sound strength of up to 144 MPa and degradation rate of 0.25 mm/a, the new redox alloying technique avoids the usage of any greenhouse active $SF_6$ gas (global warming potential 22,800) during alloying, keeping the earth's atmosphere safer. Therefore, it can be concluded that Ca-containing ceramics and hybrids are attractive alternatives to obtain, comparable to Mg-based materials, thus enabling safer processing.

**Keywords:** magnesium; synchrotron; metal injection moulding; MIM; biodegradable

## 1. Introduction

The suitability of calcium (Ca) as an appropriate alloying element for biodegradable Mg-alloys was well investigated within the last two decades [1–6]. Since a couple of years ago, in addition to common casting and machining technologies, powder metallurgy approaches based on sintering have become available. This allows processing by metal injection moulding (MIM) or additive manufacturing technologies, which are very attractive with regard to freedom in geometry as well as in costs. However, the production of the base alloys by casting is still necessary. For conventional Mg-alloying techniques, as there are melting and casting, $SF_6$ inert gas (global warming potential 22,800) is needed as a protective agent. This means 1 kg $SF_6$ is equivalent to 22.8 t of $CO_2$. In particular, for biomedical applications, rather low amounts of alloys in terms of mass but high in the number of variations in composition are needed. This makes the production of the raw material rather ineffective and the availability of specific alloy powders difficult and expensive. Alloy formation by sintering of elemental powders, or elemental powders and master alloy powders (MAP) is therefore very attractive. Thus, this study uses and translates common knowledge into a novel powder metallurgical (PM) processing method, redox-sintering. The usage of ultrafine calcium oxide (CaO) and calcium hydride ($CaH_2$) powders instead of expensive gas atomized Mg-0.8Ca alloy powder, or powder blends containing Ca-rich master-alloy [7,8], also enables homogeneous distribution of Ca into the Mg matrix. Within the chosen PM processing, no $SF_6$ inert gas is needed during

the total performance, as there is blending, alloying, and sintering, keeping the earth's atmosphere intact. This study points out how the added elementals CaO and $CaH_2$ are able to desorb at a single attendance or redox-react at a combined attendance into metallic Ca, forming the final MgCa-alloy until sintering temperature is achieved. The chosen final Ca concentration in the Mg-alloy was between 0.4 wt% and 0.8 wt%, as well as up to 8 wt% for the synchrotron experiments, enabling visibility of the Ca-phases in the in situ X-ray synchrotron diffraction pattern. The reaction mechanism is driven by Gibbs free energy relations but has never been used for alloy design in binder-based sintering techniques before [9–12]. Metal injection moulding (MIM) is a binder-based economic near-net shape prototyping technique for the production of complex-shaped parts in a high number and high reproducibility [13–15]. The scope of this work is finally focused on near-net shape 3D printing of biodegradable Mg-implant demonstrator parts using binder-based sintering techniques [16,17]. Furthermore, MIM might be useful for the manufacturing of complex-shaped biodegradable implant parts, too. Nevertheless, the form and shape of these parts cannot be spontaneously variegated because an expensive mould is needed for MIM. Hence, a logical preliminary step is to use the MIM technique as a "gold standard" for the development of novel feedstock systems to produce the tensile- and biodegradation test specimen needed within this study. Later on, the generated novel knowledge in MIM processing of these new systems shall be translated into the binder-based 3D printing technology. This novel technique did not require any mould. Hence, the idea of rapid prototyping of patient-adapted individual Mg-based biodegradable implants can be moved forward. In doing so, MIM of novel MgCa-alloy systems was carried out obtaining both sufficient material properties and biodegradation performance.

## 2. Materials and Methods

### 2.1. Powder: Production and Handling

To avoid oxygen uptake of the highly oxygen-affine powder components, the complete powder handling of particulate components was performed under a protective argon atmosphere in a glovebox system (Unilab, MBraun, Garching, Germany). Pure Mg-powder, spherical in shape, and of a size smaller than 45 μm, was used as a base material (Atoultra325, SFM, Martigny, Switzerland). For the preparation of the MgCa-blends, CaO (99.995%, 229539-5G, Sigma Aldrich) and $CaH_2$ (99.99%, 497355-10G, Sigma Aldrich) were used, considering the molar mass of calcium in the chemical compounds. Details of used Mg- and Ca-containing compounds and the MgCa-blend preparation are shown in the following Tables 1 and 2. The abbreviation ending of the blend name (_oxy/_hyd/_stoi/_ref) denotes the route used to generate the Mg-0.8Ca powder blend in the first experimental setup. Spherical and of a size smaller than 63 μm gas atomized Mg-5Ca and Mg-10Ca master alloy powders (MAP) (ZfW, Clausthal, Germany) were used to prepare a Mg-0.8Ca reference bulk material (_ref). An additional MIM processed pure Mg-feedstock was used as the control.

**Table 1.** Used powder components for blending of Mg-0.8Ca.

| Mg-Compound (g) | Ca-Compound (g) | Blend | Route |
|:---:|:---:|:---:|:---:|
| pure Mg (275 g) | CaO (3.125 g) | Mg-0.8Ca_oxy | oxide |
| pure Mg (275 g) | $CaH_2$ (2.325 g) | Mg-0.8Ca_hyd | hydride |
| pure Mg (275 g) | CaO (1.55 g) + $CaH_2$ (1.16 g) | Mg-0.8Ca_stoi | stoichiometic |
| pure Mg (260.5 g) | Mg-5Ca (46.0 g) | Mg-0.8Ca_ref | reference |
| pure Mg | - | Mg | control |

**Table 2.** Used powder components for blending of Mg-0.4Ca and Mg-0.6Ca.

| Mg-Compound (g) | Ca-Compound (g) | Blend | Route |
|---|---|---|---|
| pure Mg (275 g) | CaO (1.563 g) | Mg-0.4Ca_oxy | oxide |
| pure Mg (265 g) | Mg-10Ca (11.5 g) | Mg-0.4Ca_ref | reference |
| pure Mg (275 g) | CaO (2.35 g) | Mg-0.6Ca_oxy | oxide |
| pure Mg (270.5 g) | Mg-10Ca (17.5 g) | Mg-0.6Ca_ref | reference |

In a second, refined experimental setup, Mg-0.4Ca and Mg-0.6Ca were blended using CaO for the oxide route (_oxy) and Mg-10Ca MAP as a reference route (_ref). The Mg-10Ca (MAP) was chosen to set the later immersion test result comparable to a former study [18], which used a binder-free press and sinter technique, but identical powders.

CaO and $CaH_2$ powders obtain a sub-micro powder size, as shown in Figure 1a,b. As mentioned before, the high number of small Ca-containing particles generally enables a homogeneous distribution in the later MgCa-compound. However, ultrafine CaO and $CaH_2$ powders tend to form agglomerates, as pointed out in Figure 1a,b. For the homogenization of these agglomerates, they were separately given into a laboratory mortar inside the glovebox system, stirring the material in cyclohexane solvent for 20 min, as shown in Figure 1c.

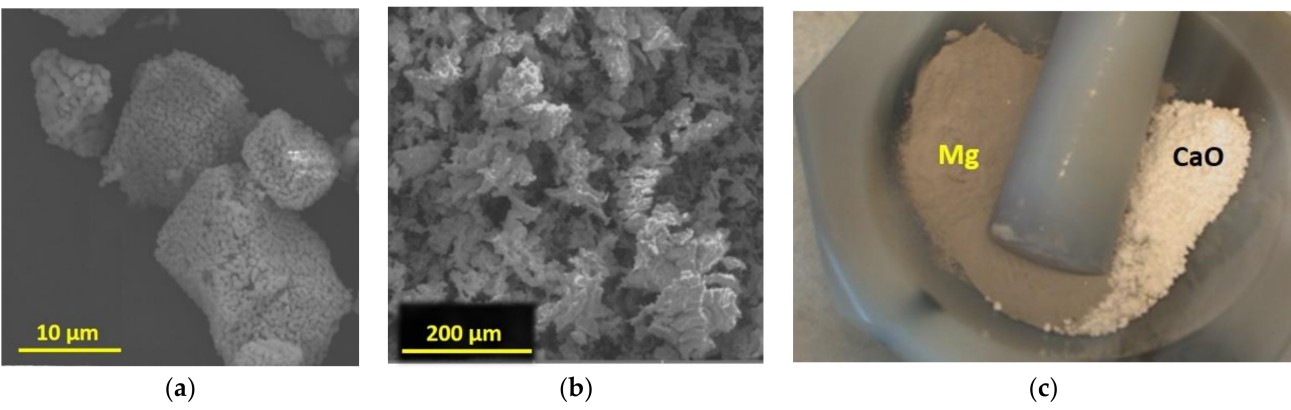

(**a**)          (**b**)          (**c**)

**Figure 1.** (**a**) SEM-image of CaO. (**b**) SEM-image of $CaH_2$. (**c**) Blending and stirring of compounds in a laboratory mortar.

Before any powder handling, feedstock preparation, or sintering took place, the used spherical raw powders were analysed in terms of corrosion supporting impurities, as there is Fe, Cu, Ni using inductively coupled plasma-optical emission spectroscopy (ICP-OES, Acros II FHX22, SPECTRO Analytical Instruments GmbH, Kleve, Germany).

*2.2. Feedstock Preparation*

Paraffin wax, stearic acid, and polypropylene-copolymer-polyethylene binder components, as shown in Table 3, were used to prepare the feedstock for the injection moulding process.

**Table 3.** Used binder components for MIM of MgCa alloys.

| Binder Component (wt%) | Abbreviation | Manufacturer |
|---|---|---|
| paraffin wax (50 wt%) | PW 58 | Merck |
| paraffin wax (10 wt%) | PW 57 | Merck |
| stearic acid (5 wt%) | StA | Merck |
| Polypropylene-copolymer-polyethylene (35 wt%) | PPcoPE | 1 |

[1] Manufacturer cannot be named due to proprietary interest.

The powder components and the organic binder components were placed in PTFE-lined stainless steel mixing beakers (see Figure 2a), preheated up to 175 °C in a furnace (Memmert, UF30plus, Schwabach, Germany), and mixed in a planetary mixer by stirring the beakers for five minutes using a 2000 rpm rotation speed (Thinky ARE 250 planetary mixer, Tokyo, Japan).

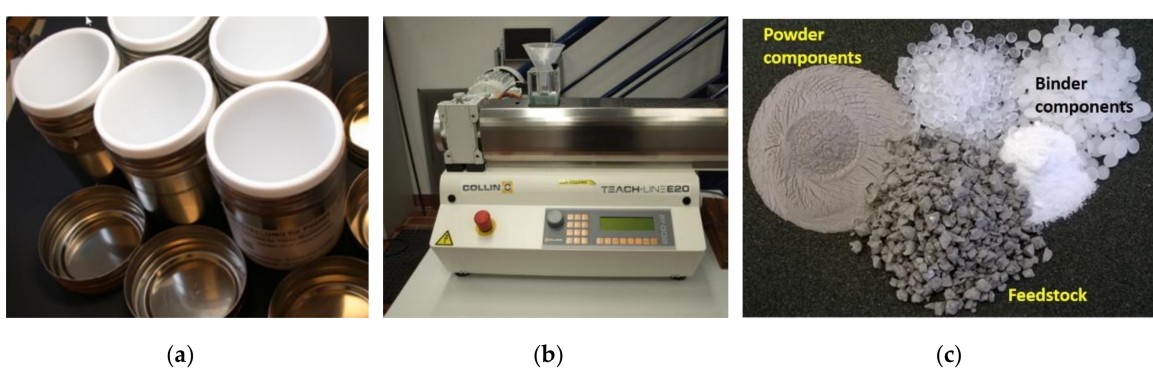

(**a**)　　　　　　　　　　(**b**)　　　　　　　　　　(**c**)

**Figure 2.** (**a**) PTFE-lined stainless steel mixing beakers. (**b**) Extruder for feedstock homogenization. (**c**) MgCa Feedstock granules showing 64 vol.% powder loading.

To achieve full homogeneity of the feedstock, the bulk material was subsequently granulated (Wanner, B08.10f, Vlotho, Germany), further on, extruded (Collin, Teach Line E20, Maitenbeth, Germany) (see Figure 2b), and once again, granulated. The powder loading of all used feedstock systems was 64 vol.%. The single components and final feedstock granules can be seen in Figures 1c and 2c.

### 2.3. Metal Injection Moulding (MIM)

The homogenous feedstock granules were injection moulded using an industrial injection-moulding machine (Arburg, Allrounder 370A, Germany) to produce the dogbone shaped tensile test specimens, as well as the cylindrical biodegradation test specimen. The tensile test specimen (l = 90 mm, d = 5.0 mm, m = 3.5 g) were consistent to ISO 2740, as shown later on in Chapter 3.1. The injection moulded biodegradation test specimens obtained were 10 mm in diameter and 50 mm in length.

### 2.4. Debinding and Sintering

Solvent debinding of all produced green parts took place in cyclo-hexane (98%, Bernd Kraft GmbH, Duisburg, Germany) at 45 °C for 15 h to remove paraffin wax and stearic acid (Lömi EBA50/2006, Großostheim, Germany). After solvent debinding, the debinded green parts were stored in the glovebox system to prevent any further oxygen uptake. Thermal debinding and sintering were maintained in a combined debinding and sintering hot-wall tube furnace with an external binder precipitation zone (MUT Advanced Heating GmbH, RRO 350-900, Jena, Germany). The furnace provides a temperature accuracy of ±1 °C using eight separately settable heating zones. Before starting any sintering process, preliminary leakage checks were done. The leakage rate at any furnace run was below $4 \times 10^{-4}$ mbar/L·s. The thermal debinding was accomplished according to the diagram

in Figure 3a. After several steps of furnace evaporation and gas purging, the furnace was heated up to 380 °C at 950 mbar in an argon atmosphere (Ar6.0) to enable fast warming of the specimen inside the inner crucible. The labyrinth-like crucible setup, getter usage, and sintering parameters were discussed in detail elsewhere [10]. Thermal debinding took place under alternating pressure at 20 mbar with a hysteresis of ±10 mbar at a 0.4 L/min argon gas flow (Ar6.0). Because Mg has the highest vapour pressure of all technical metals, it cannot be sintered under technical or even medium vacuum. This would result in evaporation of the material and further deposition in the cold binder precipitation area or even colder areas of the furnace (see Figure 3b). Sintering was conducted for 64 h at 636–646 °C furnace temperature in an argon atmosphere (Ar6.0) at 230 mbar.

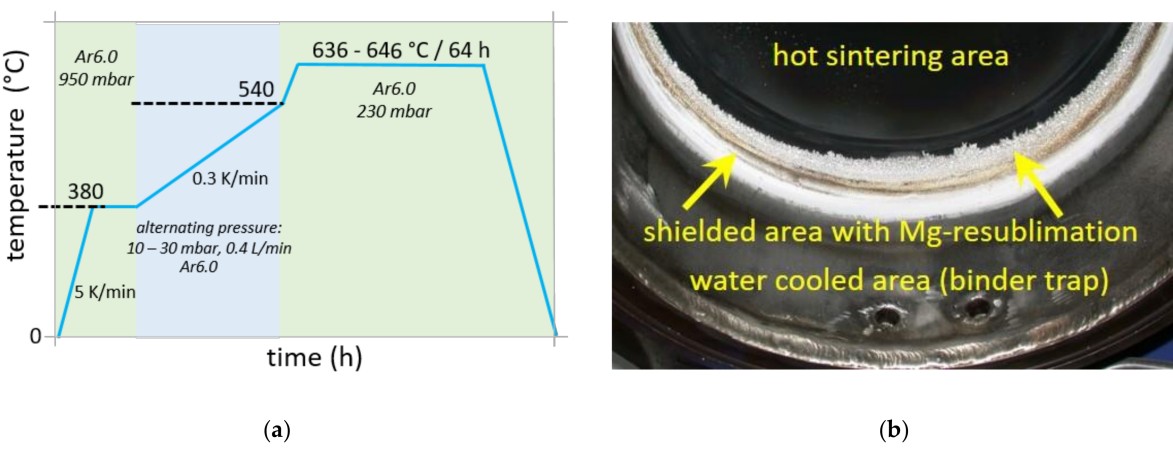

(**a**)   (**b**)

**Figure 3.** (**a**) Schematic diagram of the sintering procedure for MIM MgCa parts. The quality of the used argon gas was Ar6.0 (99.9999%). (**b**) View inside of the furnace tube showing Mg-resublimation between the hot furnace zone and the water-cooled area with gas in- and outlet.

### 2.5. Preparation of Biodegradation Test Specimen

The sintered biodegradation test specimen cylinders were cut into 2 mm thin disks avoiding any iron, copper, or nickel-containing tools or equipment (Bühler, Isomet1000, Leinfelden-Echterdingen, Germany) using a 0.5 mm SiC cutting blade (Struers, 10S15-C150TB, Willich, Germany). After the cutting of each platelet, it was directly wet ground all-round onto 2500SiC grinding paper (ATM, Saphir 360, Mammelzen, Germany) subsequently with ethanol rinsing and cleaning, following up with vacuum drying.

### 2.6. Immersion Test Setup

The immersion test was performed under semi-static conditions for 7 days in an incubator using 2 mL DMEM (Dulbecco's Modified Eagle Medium) þ GlutaMAX™ þ 10% FBS (Fetal Bovine Serum) solution under cell culture conditions (37 °C, 5% $CO_2$, 19.7% $O_2$, 95% relative humidity). The immersion test was set up in 24 multi-well plates (Thermo Fisher Scientific, Hvidovre, Denmark) under a sterile environment. The immersion medium was changed every 2–3 days. The degradation rate was calculated by the measurement of weight loss after the removal of degradation products by chromic acid [19].

### 2.7. Characterisation Methods

Geometrical data and mass of sintered dog-bone shape tensile test specimen and cylindrical biodegradation test specimen were measured in the as green and as-sintered condition using Equation (1) to evaluate shrinkage $s_f$ and Equation (2) to evaluate total residual porosity $P_x$. Therefore, a balance (A&D, FZ-300i, Oxford, UK) and a capiller (Mahr-16EX, Wuppertal, Germany) were used. Regarding Equation (1), $l$ is the specimen length in the as green ($l_g$) and as-sintered ($l_s$) condition. Regarding Equation (2), $m_s$, $l_s$,

and $d_s$ are mass, length, and diameter in the as-sintered condition and $\rho_{th}$ is the theoretical density of the MgCa alloy ($\rho_{th}$ = 1.738 g/cm$^3$).

$$sf = 1 - \left( \frac{ls}{lg} \right) \tag{1}$$

$$Px = 1 - \left( \frac{\frac{ms}{ls*\Pi*\frac{ds2}{4000}}}{\rho th} \right) \tag{2}$$

The tensile testing took place using a material's testing machine (Zwick/Roell, Retroline Z050, Ulm, Germany). The microstructure was observed using light microscopy (Olympus, DSX, Hamburg, Germany) as well as scanning electron microscopy (SEM) equipped with energy dispersive X-ray (EDX) (Tescan Vega3, operating at 15 to 17 kV in the BSE-mode, Brünn, Czech Republic). For the investigation of phase formation, compound desorption, and redox reactions in situ X-ray synchrotron radiation analysis was performed with a photon energy of 100 keV. The diffraction patterns were recorded every ~30 s using a Perkin Elmer XRD 1622 Flatpanel detector (DESY, PETRA III, HEMS beamline P07, Hamburg, Germany).

## 3. Results and Discussion

### 3.1. Shrinkage and Residual Porosity

Table 4 summarises the sintering results of redox-alloyed and -sintered Mg-0.8Ca specimens using calcium oxide (_oxy), calcium hydride (_hyd), and a stoichiometric mixture from both (_stoi). This first specimen setup consisted only of binary Mg-0.8Ca alloys. The only difference between these four blended Mg-0.8Ca routes (_oxy, _hyd, _stoi, _ref) is how the way calcium was introduced into the alloy. This technique was described in detail in Section 2.1. The procedure of introducing the Ca into the alloy determined the differences in shrinkage behaviour and residual porosity of the samples produced by the four blended Mg-0.8Ca routes. The sintering results of the reference route (_ref) and a Ca-free control route, using the pure Mg, were also included in Table 4. The sintering mechanism of the pure Mg control material is a pure solid phase diffusion process without the help of an initial liquid phase. Hence, if one of the Ca-containing routes shall not be able to form an initial Ca-rich liquid phase, it might show the same sintering performance as that of the pure Mg control material.

**Table 4.** Shrinkage $_{sf}$ and residual porosity $P_x$ of sintered Mg-0.8Ca blends, sintered at 646 °C for 64 h. The number of specimens per set was $n$ = 4.

| Sintered Mg-Blend | Shrinkage s$_f$ | Residual Porosity P$_x$ | Route |
|---|---|---|---|
| Mg-0.8Ca_oxy | 12.0% | 2.1% ± 1.4% | oxide |
| Mg-0.8Ca_hyd | 8.9% | 3.3% ± 1.9% | hydride |
| Mg-0.8Ca_stoi | 11.6% | 3.0% ± 1.0% | stoichimetric |
| Mg-0.8Ca_ref | 11.0% | 1.8% ± 1.0% | reference |
| pure Mg | 4.1% | 22% ± 0.5% | control |

Table 4 determined that shrinkage and residual porosity of all chosen redox alloyed routes, as well as the reference route, pointed out sufficient sintering performance showing typical low residual porosity of the MIM-parts and high shrinkage, close to the theoretical maximum. In comparison, the pure Mg control material suffers from 22% remaining porosity at 4.1% shrinkage. The following Figure 4 visualizers the exemplary shrinkage of the sintered tensile test specimen of the redox-alloyed materials and the pure Mg control material to their corresponding green parts. The shrinkage differences between

the sufficient sintered redox alloyed materials and the control material can be visualised directly.

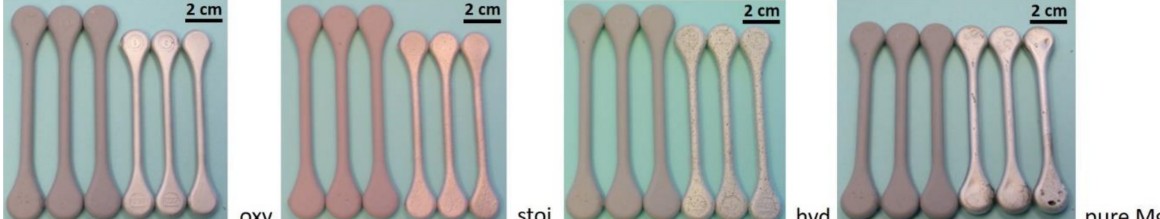

**Figure 4.** Dog-bone shape tensile test specimen produced by MIM of Mg-0.8Ca feedstock The left side parts are in the green condition as injection moulded; right side parts are the corresponding parts in the as-sintered condition. Shrinkage of the as-sintered parts in comparison to the corresponding green parts can be seen very well.

### 3.2. In Situ Synchrotron X-ray Diffraction

The result shown in the previous chapter gives a hint that the assumed initial solid-state diffusion reaction between solid Mg and solid CaO, $CaH_2$, respectively, already took place at temperatures where the single components (Mg, CaO, $CaH_2$) remained in a solid phase, but at cooperative attendance with Mg, forming an initial MgCa-liquid phase at the particle's interface.

The sintering results, shown in Section 3.1, can be underlined by in situ X-ray synchrotron diffraction analysis of the chosen alloying routes using 8 wt% Ca instead of 0.8 wt%, to enable better visibility of the added calcium oxide- and calcium hydride-lines, as well as the formed secondary phases in the X-ray diffraction pattern.

Figure 5 shows in situ X-ray diffraction patterns of the chosen redox alloyed powder blends, as well as of the reference powder blend. The continuous lines at around 3.5° 2-Theta and 4° 2-Theta show high intensity originating from the steel crucible, used for the experiments. The three blueish lines below 3° 2-Theta can be attributed to the hexagonal crystal structure of the Mg-material. These lines are interrupted in the middle of the diagrams at a temperature around the sintering temperature due to the formation of a liquid amorphous phase. The chosen maximum sintering temperature was 640 °C, dwelled for five minutes. The shown patterns point out the occurrence of an initial liquid phase independent of the chosen alloying route, as visible throughout the amorphous phase region at temperatures close to sintering temperature. Hence, it can be presented that at the mutual attendance of pure Mg-powder and calcium oxide, calcium hydride, or both elementals, an initial solid-phase diffusion reaction between Mg and these elementals takes place, forming a liquid sintering initiating phase, despite the fact that neither pure Mg nor the chosen elementals would form any liquid phase at single attendance at the chosen temperature.

The following in situ X-ray diffraction pattern in Figure 6 points out the phase transformations during the described heat treatment in higher resolution, exemplary for the Mg + CaO system.

Both the diffraction pattern of added pure Mg powder as well as pure CaO-powder can be seen in the lower region of the diagram where the heat treatment started at room temperature (see red and blue arrows). At temperatures above 628 °C, the blend initially started to form a fully liquid amorphous phase. A closer look at this area (see the orange square on the right side) clarified that first of all, the CaO pattern disappears at around 620 °C, forming the first initial amorphous liquid (see the grey shade area inside of the orange square). Around 628 °C, the complete or full melting of the components occurred. During this process, the material blend formed its equilibrium alloy according to the added amounts of elements. Below 628 °C, in the cooling segment of the heat treatment, the material began to crystalize again. Hence, 628 °C seems to be approximately the liquidus temperature of the equilibrium alloy. Below the eutectic temperature of around 450 °C,

fully crystallisation took place again, forming a new X-ray diffraction pattern, as there is the intermetallic phase $Mg_2Ca$ (see green arrows). In return, the intensity of the CaO pattern decreases (see blue arrows in the upper region of the diagram), or even disappears fully, respectively. Some patterns that occur unmodified through the diagram are corresponding to the used stainless steel crucible (see black arrows). Disadvantageously, the occurrence of MgO phases (purple arrows) cannot be identified clearly due to the fact that the main MgO patterns are overlaid by the steel pattern of high intensity at the same positions (see black arrows) at 3.94° and 6.55° of 2-Theta angle. However, the demonstrated results gave a hint that the following reaction took place:

$$3Mg + CaO \rightarrow Mg_2Ca + MgO \tag{3}$$

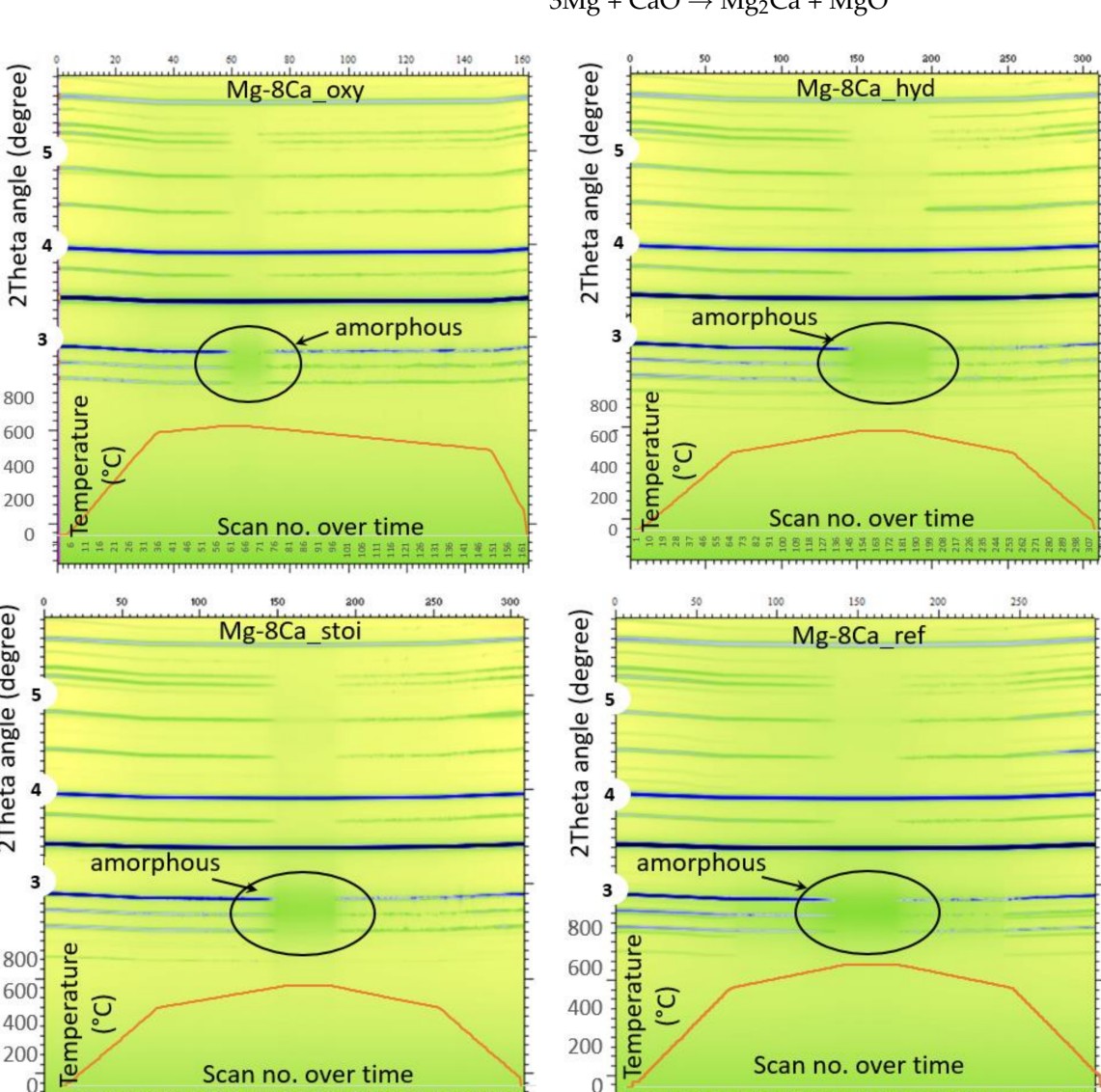

**Figure 5.** In situ X-ray diffraction pattern of Mg-8Ca materials showing amorphous area.

where an increase of MgO cannot be clearly detected in the in situ XRD pattern, using the chosen experimental setup. The $Mg_2Ca$-phase, shown in Figure 6 (green arrows), cannot be formed without the as shown reactions taking place. The diagram figures out that the steel pattern at 3.94° 2-Theta splits into two visible patterns during the process, ending up at 3.94° 2-Theta and 4.02° 2-Theta. This phenomenon can also be observed using an empty steel crucible without any Mg-specimen inside as a control measurement. Hence, the observed splitting of the pattern cannot be interpreted as MgO formation. According to

established literature, in binary systems, the standard Gibbs free enthalpy of formation ΔG is more negative for CaO than that of MgO, and therefore, CaO shall be more stable than MgO [9,10]. However, this shall be valid at low temperatures only. Recent research pointed out the dependency of temperature and ratio of ingredients, which enables the clockwise reaction, as shown in Equation (1), too [11,12].

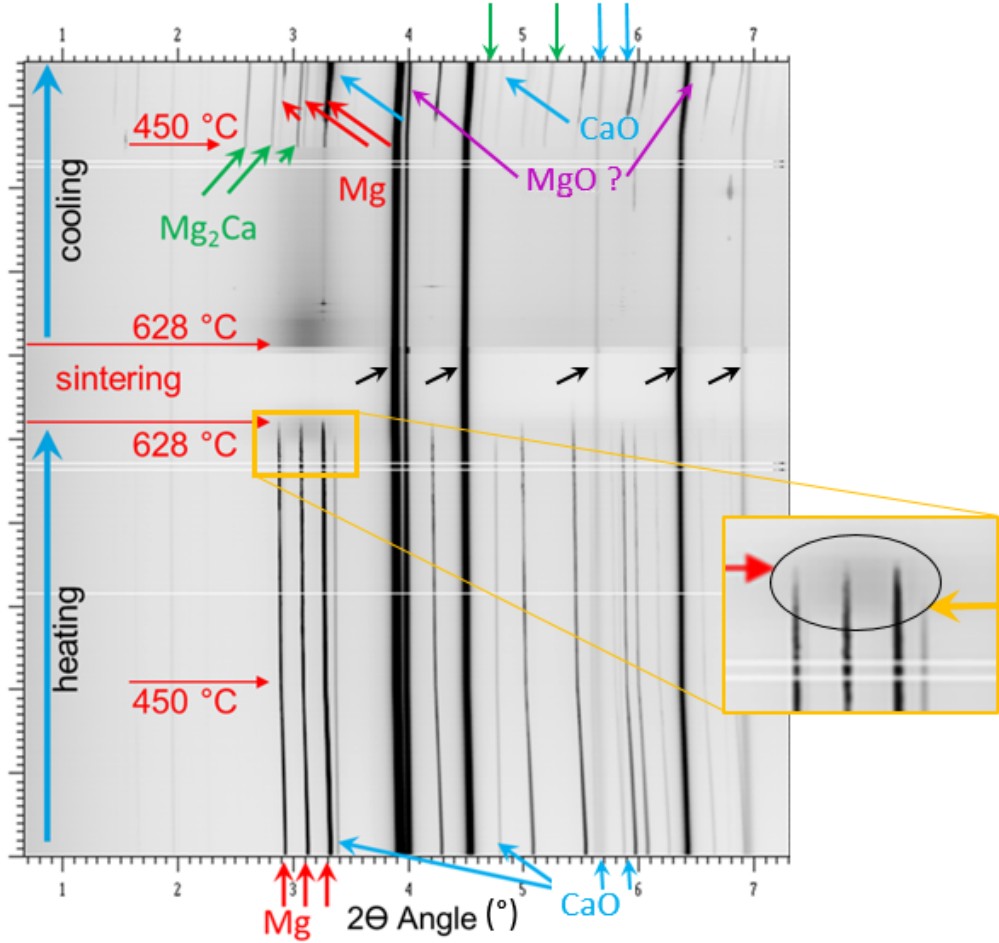

**Figure 6.** In situ X-ray diffraction pattern of powder blend: Mg + 2 vol.% CaO showing amorphous area and phase transformations. The *y*-axis directly corresponds to the XRD-scan number which indirectly corresponds to the temperature of the specimen inside of the dilatometer furnace (independent of the furnace program). Hence, some meaningful temperature values are given in the diagram. Blue arrows on the upper and lower *x*-axis correspond to the CaO-patterns. Red arrows correspond to Mg, green arrows correspond to the intermetallic phase Mg₂Ca, black arrows correspond to the patterns of the used stainless steel crucible. The steel patterns are not interrupted. Purple arrows correspond to MgO, whereas the increase of MgO cannot be detected clearly due to the overlay of the crucible pattern (steel).

### 3.3. Mechanical Properties and Microstructure

The tensile test results of as-sintered Mg-0.8Ca MIM-specimens are shown in the following diagrams in Figures 7 and 8. For a better result assessment in view of the chosen sintering parameters, lower alloyed Mg-0.6Ca_ref and Mg-0.4Ca_ref reference materials are additionally integrated into the diagrams. Figure 7 points out that at the sintering temperature of 642 °C, the best redox-sintering result could be achieved using MgO as an elemental alloying element, obtaining an ultimate tensile strength (UTS) of 144 MPa at 6.6% elongation at fracture. The maximum UTS of 153 MPa at 7.7% elongation at fracture could be achieved using the Mg-0.8Ca_ref reference material consisting of pure Mg and Mg-5Ca (MAP) as a calcium source. Surprisingly, the lower alloyed Mg-0.4Ca_ref material

could achieve 154 MPa at 7.5% elongation at fracture. This is a useful result for further biomedical application because of reduced material production costs and a lower body burden of unnecessary foreign element loads.

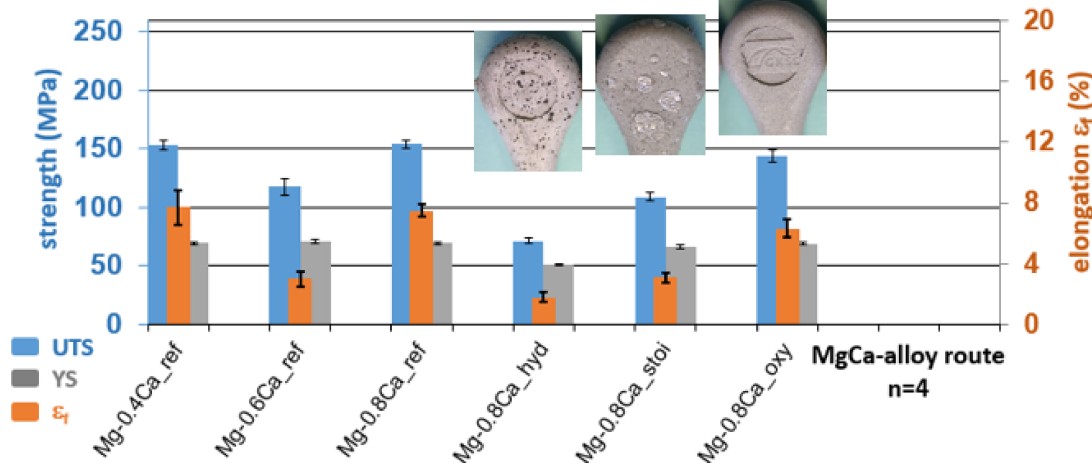

**Figure 7.** Tensile test results of the MgCa materials, sintered at 642 °C for 64 h showing UTS (blue), yield strength (grey), and elongation at fracture $\varepsilon_f$ (orange).

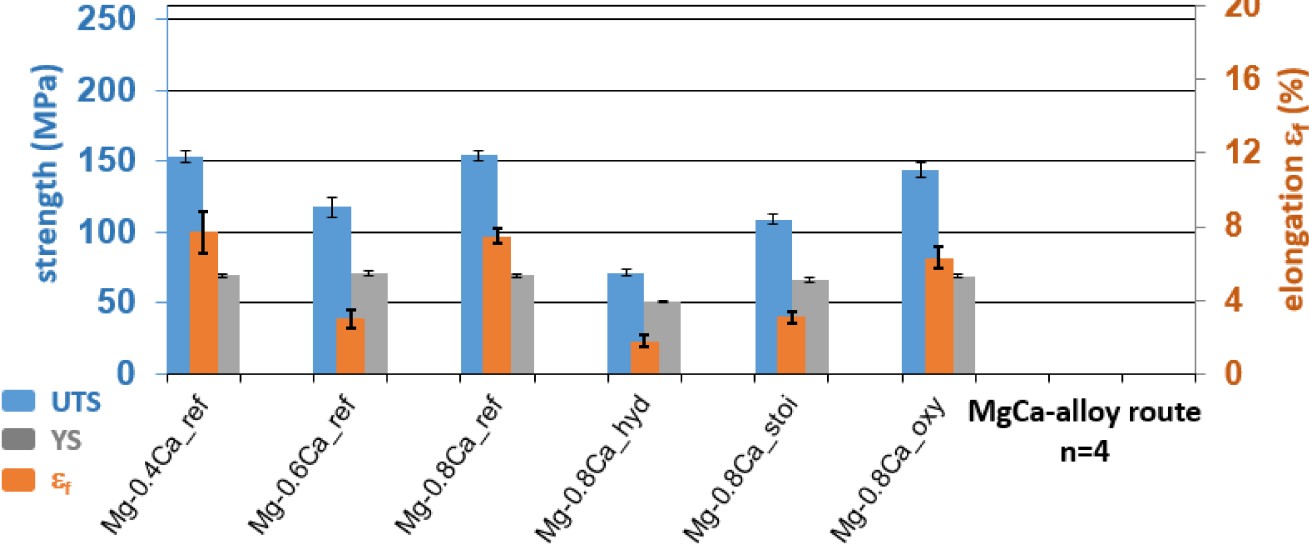

**Figure 8.** Tensile test results of the MgCa materials, sintered at 646 °C for 64 h showing UTS (blue), yield strength (grey), and elongation at fracture $\varepsilon_f$ (orange).

The three images of the redox-alloyed specimens (integrated in Figure 7) show their surface after sintering. The Mg-0.8Ca_oxy oxide route obtained a perfectly spotless surface. In contrast, the Mg-0.8Ca_hyd hydride-route suffered from coarse pores, which might be the main reason for the decrease in strength. The Mg-0.8Ca_stoi material did not show these described porosities despite the fact that the material contained calcium hydride, too. However, the stoichiometric route shows surface coarsening. This observation indicated that the optimal sintering temperature of the material was gained or even exceeded.

Because the results shown in Figure 7 are valid for one sintering temperature, only, further setups at different sintering temperatures are needed to get a useful overview of the sintering performance of the chosen blends and alloys. The following diagram in Figure 8 shows the results of the same alloys at 646 °C sintering temperature. Again, it can be seen that the Mg-0.4Ca_ref material achieves the maximum material properties of 153MPa (UTS)

at 7.1% elongation at fracture. The Mg-0.8Ca_hyd material could increase its properties in comparison to the former run at 642 °C.

For the use of $CaH_2$ as an elemental sintering agent, the following result can be seen: On the one hand, calcium hydride desorbs into hydrogen and calcium, forming a calcium-rich liquid MgCa phase, which initiates a temporary liquid phase sintering process, as shown in the former chapter and in the following micrographs in Figure 9. Hence, a moderate strength can be achieved. On the other hand, the desorption of solid calcium hydride into gaseous hydrogen seemed to be performed very quickly, affecting the formation of coarse pores inside the microstructure, as shown in the specimen images, integrated in Figure 7, and in the microstructural image in Figure 9.

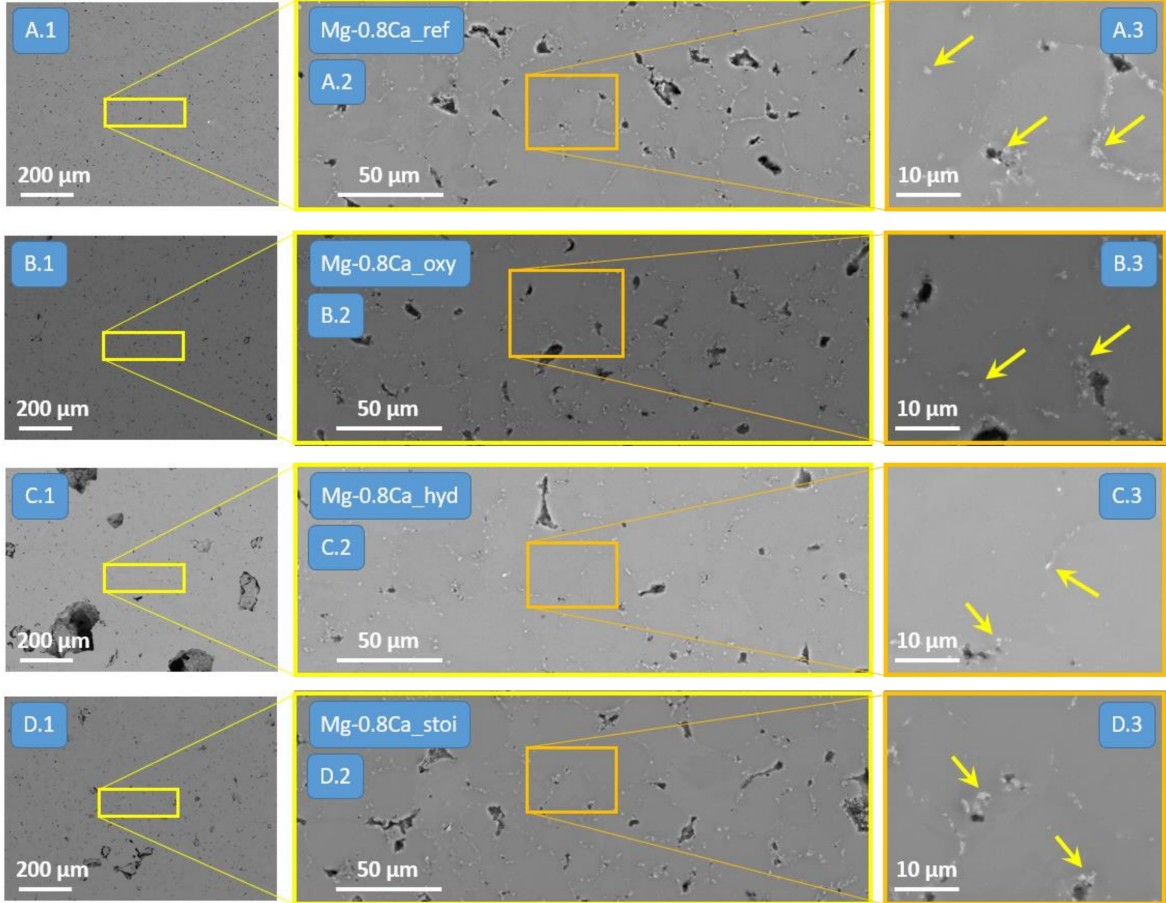

**Figure 9.** SEM images in BSE-mode, using 17 kV at 10 mm working distance. Images are showing the microstructure of the Mg-0.8Ca materials sintered at 646 °C for 64 h. Right-hand side (**A.1**–**D.1**): overview-image of macroscopic microstructure at low magnification (250×). Squares are indicating the area of the higher magnified image. Middle (**A.2**–**D.2**): microstructure and porosity at 1000× magnification. Left-hand side (**A.3**–**D.3**): grain boundaries and secondary phases (see yellowish arrows) at 5000× magnification.

Figure 9 points out both aspects of $CaH_2$ as a sintering agent very descriptively, where two kinds of porosity can be seen: On the one hand, generally very good sintering bounding and low residual porosity, shown in images C.2 and C.3 of Figure 9. On the other hand, a coarse macroscopic porosity, which surrounds the fine structures and good sintering performance, can be seen in image C.1. These pores are reducing the materials' strength and stiffness. Therefore, a better way of homogenization of calcium hydride in the powder blend has to be found in further work.

The stoichiometric route (_stoi) is able to form $H_2O$ as a reaction product without any balance of gaseous oxygen or gaseous hydrogen. The micrographs for the stoichiometric

route in Figure 9D.1–D.3 point out a significant reduction of the formation of a coarse, macroscopic porosity, as described at the usage of calcium hydride, only. Hence, lower residual porosity, higher shrinkage (see Table 4), and higher strength (see Figure 7) can be observed. Additionally, the specimen image integrated in Figure 7 shows that the stoichiometric route achieved or even exceeded its optimal sintering temperature, forming coarse-grained structures onto the surface. Hence, it was not surprising that a further increase of sinter temperature resulted in a decrease of the UTS and elongation at fracture, as shown in Figure 8.

Best redox-sintering results could be found using calcium oxide as an elemental alloying element performing a smooth specimen surface and highest strength, comparable to that of the reference material, as shown in Figures 7 and 8, and the micrographs B.1–B.3 in Figure 9. The calcium oxide and calcium hydride particles, as shown in Figure 1a,b could not be found anywhere in the as-sintered materials. Images A.3, B.3, C.3, and D.3 in Figure 9 illustrate bright secondary phases onto the grain boundaries (see yellowish arrows). EDX analysis visualised slightly increased calcium and oxygen concentrations within these phases in comparison to the greyish Mg matrix. Indicating the formation of $Mg_2Ca$, MgO and CaO or the non-equilibrium reaction between the phases (Mg) + $Mg_2Ca$ + MgO + CaO [11,12]. The EDX result could be detected in any used alloy, independent of the chosen alloying route.

The diagrams in Figures 7 and 8 point out that the Mg-0.4Ca_ref alloy seemed to be the most temperature-tolerant alloy composition. In contrast, the Mg-0.6Ca_ref material expressed the lowest mechanical properties of the reference materials. It was assumed that the Mg-0.6Ca properties shall be between the Mg-0.4Ca and the Mg-0.8Ca results, but this is not the case. Because of these open questions and the fact that the Mg-0.8Ca alloys seemed to form too much liquid phase at 646 °C, the Mg-0.4Ca, and the Mg-0.6Ca material were chosen again for the second setup to verify the former result or to verify the assumption. For the second specimen setup, fully new feedstocks were produced under extra clean conditions. Especially with respect to the overall reliability and repeatability, the feedstocks were produced using fully metal-free tools and crucibles to avoid any iron contamination. The compositions shown in Table 2 were chosen. In view of the immersion tests, the results shall be comparable to a formerly mentioned PhD study using Mg-0.6Ca alloy and an Mg-10Ca MAP [18]. In that study, an identical blend was used under binder-free press and sinter conditions.

Figure 10 images E.1–F.3 prove the assumption about the sintering performance of the Mg-0.6Ca_ref material in the former chapter. The new Mg-0.6Ca_ref material obtain 1.4% residual porosity followed by the Mg-0.6Ca_oxy oxide route achieving 2% residual porosity. This result indicates that the 646 °C sintering temperature is corresponding very well with the 0.6wt% Ca amount in the binary MgCa alloy, forming an optimal amount of sintering initiating the Ca-rich liquid phase, according to the liquidus temperature of the formed equilibrium. In contrast, the sintering process of the Mg-0.4Ca_ref material seems to be marginally (4.9% residual porosity), and that of the Mg-0.4Ca_oxy material majorly (14.1% residual porosity) incomplete, as shown in image G.1–H.3 of Figure 9. This result points out that the sintering temperature, and hence, the liquidus temperature of the formed equilibrium alloy seems to be marginally too low for 0.4wt% Ca content in the binary alloy. This indicates that comparable and reproducible perfect sintering with low residual porosity might be possible using a marginally higher sintering temperature for the 0.4wt Ca-alloys. This result also indicates that perfect sintering parameters can be found on the one hand, but on the other hand, the optimal processing window for a chosen alloy, independent of 0.4wt%, 0.6wt%, or 0.8wt% Ca content, seems to be quite small.

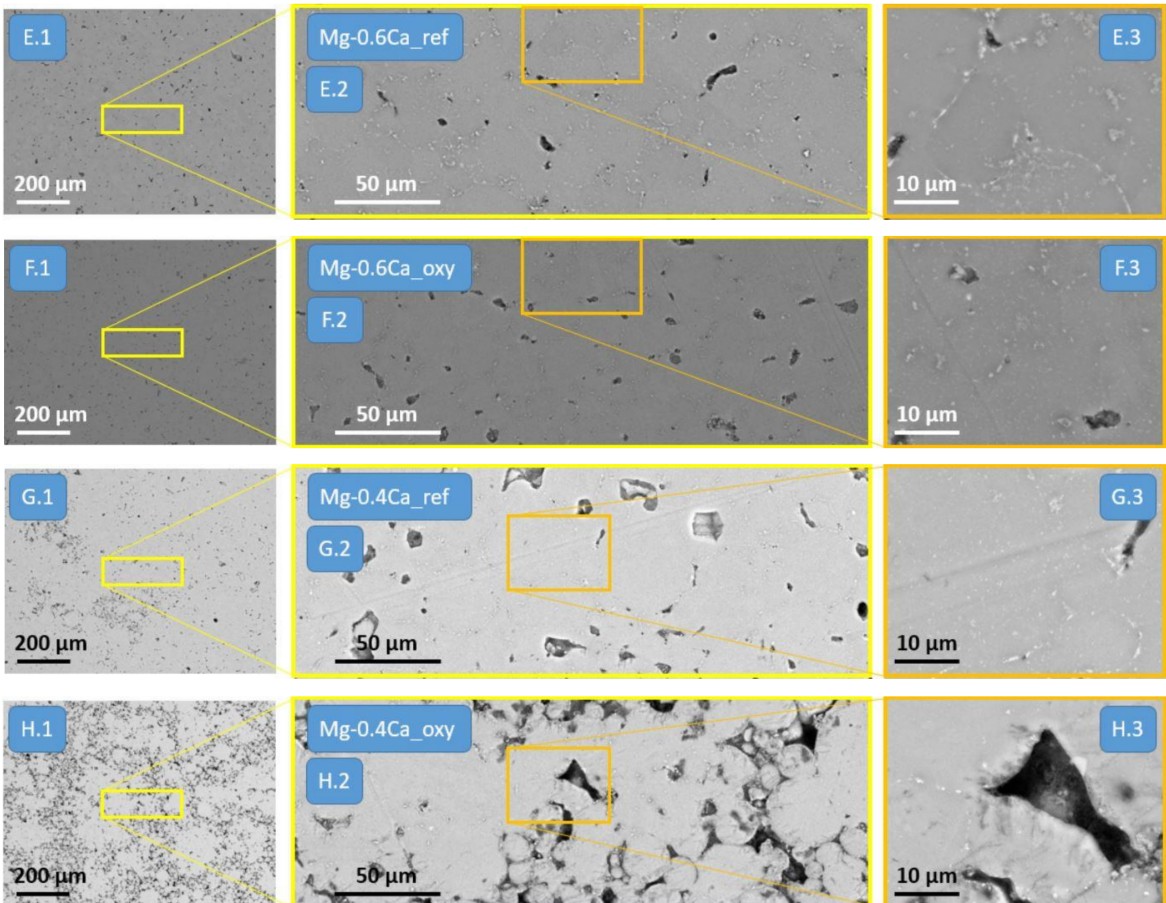

**Figure 10.** SEM images of the microstructure of the Mg-0.4Ca and Mg-0-6Ca materials sintered at 646 °C for 64 h. SEM images are performed in BSE-mode, using 17 kV at a 10 mm working distance. Right-hand side (**E.1–H.1**): overview-image of macroscopic microstructure at low magnification (250×). Squares are indicating the area of the higher magnified image. Middle (**E.2–H.2**): microstructure and porosity at 1000× magnification. Left-hand side (**E.3–H.3**): grain boundaries and secondary phases at 5000× magnification.

### 3.4. Biodegradation Test

The results of the ICP-OES measurements of the used powders are shown in the following Table 5. These results are needful for the interpretation of the immersion test results in view of MIM processing.

**Table 5.** Results of the ICP-OES measurements of the used powders in [mg/kg] (ppm).

| Powder | Fe (mg/kg) | Cu (mg/kg) | Ni (mg/kg) |
|---|---|---|---|
| Pure Mg | 39 | 3 | 3 |
| Mg-5Ca (MAP) | 15 | 10 | 3 |
| Mg-10Ca (MAP) | 21 | 13 | 2 |

The results in Table 5 point out that the iron level neither, nor copper, nor nickel concentration exceeds any corrosion relevant tolerance limits. According to common literature sources, the critical tolerance limit for iron in magnesium seems to be 50 ppm [20–22]. Accordingly, satisfying immersion test results shall be able to achieve using these powders, if no further iron, copper, or nickel uptake takes place during further MIM processing. Hence, the intentions of an extra clean powder and feedstock handling, using metal-free tools, as well as most as possible metal-free MIM processing, can be clarified. Table 6

highlights the results of the immersion tests in comparison to the residual porosity of the chosen MIM processed and sintered Mg-0.4Ca to Mg0.6Ca specimens.

**Table 6.** Results of the immersion tests using MIM processed and sintered Mg-0.4Ca to Mg-0.6Ca specimens. Number of specimens per set was *n* = 8.

| Sintered MIM-MgCa-Blend (*n* = 8) | Mean Degradation Rate (mm/a) | Residual Porosity $P_x$ (%) |
|---|---|---|
| Mg-0.4Ca_ref | 0.51 ± 0.09 | 4.9 ± 0.4 |
| Mg-0.4Ca_oxy | - | 14.1 ± 1.3 |
| Mg-0.6Ca_ref | 0.32 ± 0.16 | 1.4 ± 0.3 |
| Mg-0.6Ca_oxy | 0.25 ± 0.04 | 2.0 ± 0.3 |

Table 6 pointed out that both the Mg-0.6Ca materials as well as the Mg-0.4Ca_ref material showed sufficient degradation performance below the tolerance value of 1 mm/a. Specimen sets obtaining low residual porosity also obtain a low mean degradation rate. The mean degradation rate of the Mg-0.4Ca_oxy material could not be detected, due to the ebullition of the medium during the experiment. This can be explained by its high residual porosity of 14.1%. The results in Table 6 are consistent with the literature, which reveals a residual porosity of above 7% as a critical value for the mean degradation rate. This can be explained by the fact that above 7%, the residual porosity changes from closed porosity to open porosity. Hence, the medium can intrude into the specimen, accelerating the degradation process immediately. In the case of open porosity, the total real surface area is much higher than the outer specimen surface only.

## 4. Conclusions

This study points out the general suitability of calcium oxide (CaO) and calcium hydride ($CaH_2$) as alternative elementals to commonly used Ca-rich master alloy powder (MAP) to form MgCa alloys, obtaining sufficient sintering, strength, and biodegradation performance. On the one hand, the utilisation of $CaH_2$ resulted in sufficient sintering performance but on the other hand in the occurrence of coarse pores inside the microstructure. This pore formation effect diminished the theoretical achievable material properties. Further studies have to be performed in order to find a process key to avoiding the formation of these macroscopic pores. A better way to de-agglomerate the nanosized $CaH_2$ powders has to be found. A stoichiometric blend of $CaH_2$ and CaO reduces the formation of these coarse pores. The best redox results could be found using CaO as a calcium source, obtaining an ultimate tensile strength (UTS) of 144 MPa at 6.6% elongation at fracture and a 0.25 mm/a mean degradation rate. Moreover, the chosen binder-based powder metallurgical (PM) blending and sintering route avoids the need for any protective $SF_6$ inert gas (global warming potential 22,800) during the entire process, keeping the earth's atmosphere intact. Further on, the shown PM-blending route eases the manufacturing of divers Mg-alloy compositions without the need of cast shop and gas atomizer, possibly interesting for smaller companies dealing with Mg-feedstock and Mg-filament fabrication for future 3D printing applications. Generally, this study demonstrated, worldwide, for the first time, the suitability of MIM processed MgCa-alloy for biomedical applications, obtaining both a sufficient mechanical strength and degradation rate.

**Author Contributions:** M.W. conceived and designed the experiments; M.L., H.H., D.S. and M.W. performed the experiments and analysed the data; H.H. contributed materials and analysis tools; M.W., H.H., T.E. and R.W.-R. wrote the paper. All authors have read and agreed to the published version of the manuscript.

**Funding:** This research received no external funding.

**Data Availability Statement:** The data is available on request from the author.

**Conflicts of Interest:** The authors declare no conflict of interest.

## Abbreviations

The following abbreviations are used in this manuscript:

| | |
|---|---|
| MIM | metal injection moulding |
| PM | powder metallurgy |
| MAP | master alloy powder |
| UTS | ultimate tensile strength |
| YS | yield strength |
| _oxy | oxide route |
| _hyd | hydride route |
| _stoi | stoichiometric route |
| _ref | reference route |
| PW 58 | paraffin wax, melting point 58 °C |
| PW 57 | paraffin wax, melting point 53 °C |
| StA | stearic acid |
| PPcoPE | polypropylene-copolymer-polyethylene |
| PTFE | polytetrafluoroethylene |
| DMEM | Dulbecco's Modified Eagle Medium |
| FBS | Fetal Bovine Serum |
| SEM | scanning electron microscopy |
| EDX | Energy Dispersive X-ray |
| DESY | Deutsches Elektronen Synchrotron |
| ICP-OES | Inductively Coupled Plasma- Optical Emission Spectroscopy |
| Ar6.0 | Argon, purity 99.99990% |

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
