# Peer review of "In Situ X-ray Synchrotron Radiation Analysis, Tensile- and Biodegradation Testing of Redox-Alloyed and Sintered MgCa-Alloy Parts Produced by Metal Injection Moulding"

_metals, doi:10.3390/met12020353_

Round 1

Reviewer 1 Report

Dear Authors.

1.How You control the temperature? The difference between temperature 642 and 646 is only 4 celsius degree.

2.Is possible to control so exactly the temperature in used furnace?

3.Is the thermocouple inside in the tube?

Generally the article is very interesting

Author Response

Dear Reviewer,

many thanks - here are my answers regarding your questions:

1.How You control the temperature? The difference between temperature 642 and 646 is only 4 celsius degree.

          The furnace, mentioned in chapter 2.4 (MUT Advanced Heating GmbH, RRO 350-900, Germany) provides a temperature accuracy of +/- 1°C and has a thermocouple (TC) inside of the tube. More exacty, the TC is in the sintering crucible and can be moved during the sintering. I added some more information in chapter 2.4

2.Is it possible to control so exactly the temperature in used furnace?

         Yes, the furnace has 8 heating zones! Any zone can be controlled separately; even a temperature offset of any zone can be performed individual.

3.Is the thermocouple inside in the tube?

          Yes.

many Thanks,

Reviewer 2 Report

The presented scope of research is very interesting and shows the great commitment of the authors to the work carried out. As I have already received the corrected version during the review, there are definitely fewer comments.

I present them below:
The introduction is written very generally, there are few references to similar studies.
and some questions:
- in chapter 2.4 argon and Ar6 appear interchangeably - are they the same atmosphere?
- Figure 6 has a caption that looks like the phenomena in the figure, please shorten the caption and describe the phenomena in the text.
- Figure 9 shows precipitate, that could be signed or precisely indicated.

best regards

reviewer 

Author Response

Dear Reviewer,

many thanks - here are my answers regarding your questions:

I present them below:
The introduction is written very generally, there are few references to similar studies.
and some questions:
- in chapter 2.4 argon and Ar6 appear interchangeably - are they the same atmosphere?

The introduction is now more specific and involves more details regarding the aim and performance of this investigation.

Yes, all sintering was performed under argon atmosphere, quality Ar6.0 as shown in figure 3a. I´ve checked the manuscript for a consistent and more clearly notation “argon atmosphere (Ar6.0)”. Chapter “Abbreviations” line 458 is explaining the notation “Ar6.0”- Figure 6 has a caption that looks like the phenomena in the figure, please shorten the caption and describe the phenomena in the text.

Possibly, you mean the word “crucible” and the black arrows within the figure, sure?. Hence, I deleted the word “crucible” and shorten the black arrows. Hope this is well. This part is now described in the text. If this is not the phenomena you mean, please refer directly to me via email (martin.wolff@hereon.de). Make a screenshot of figure 6, add this in e.g. PowerPoint, add arrows and Text inside of the figure to mention the phenomena you mean. - Figure 9 shows precipitate, that could be signed or precisely indicated.

I refined figure 9 by using additional yellowish arrows, which highlighted the precipitates and I mentioned this more precisely in the text (see line 343-348)

Many Thanks and best regards,

Martin

Reviewer 3 Report

Binary MgCa alloys are one of the promising and well investigated biodegradable metals. The manuscript is devoted to the study of a novel processing routes of MgCa alloys and avoiding the usage of any greenhouse active SF6 gas.The following suggestions should be considered when revising the manuscript:

(1)In the introduction, the research progress of MgCa alloys, including its performance achieved, should be appropriately supplemented.

(2)Table 4 lists the shrinkage and residual porosity of sintered Mg-0.8Ca blends. The shrinkage can be easily obtained by detecting the size change, but the porosity needs to be supported by metallographic photos.

Author Response

Dear Reviewer,

many thanks - here are my answers regarding your questions:

(1)In the introduction, the research progress of MgCa alloys, including its performance achieved, should be appropriately supplemented.

            The introduction is now more specific and involves more details regarding the aim and performance of this investigation.

(2)Table 4 lists the shrinkage and residual porosity of sintered Mg-0.8Ca blends. The shrinkage can be easily obtained by detecting the size change, but the porosity needs to be supported by metallographic photos.

              The residual porosity (Px) was calculated by between real density to theoretical density of the alloy using equation (2), given in chapter 2.7. The real density was calculated by measuring the geometrical data of the cylindrical specimens and the mass as described in chapter 2.7. Regarding our experiences, the usage of metallographic photos for porosity calculations differs too much between individual operators due to the possibility of differences in brightness/contrast settings during imaging and later individual settings during threshold value analysis.  

Many Thanks and best regards,

Martin

Round 2

Reviewer 3 Report

Metal powder injection molding has been developed for more than 30 years,and its advantages have been very clear.  Is it necessary to prove this point with multiple own papers?

Author Response

Dear Reviewer,

sure, as many may not be necessary. I could reduce the number of own publications. Hope you find this well.

Sure, MIM is well established for nearly 4 decades. However, the store of knowledge is only valid for the standard materials steel, non ferrous metals and titanium alloys. In contrast, sintering of magnesium was known as impossible due to a stable oxide layer onto the particles surface, inhibiting the diffusion process, necessary for sintering success. Another challenge of MIM of Mg-alloys is to find an appropriate binder system, not reaction with magnesium. The own studies, shown in the manuscript pointed out step by step how to overcome these challenges. This might help the reader to have success with own experiments.